# Physicochemical and Organoleptic Differences in Chardonnay Chilean Wines after Ethanol Reduction Practises: Pre-Fermentative Water Addition or *Metschnikowia pulcherrima*

**Candela Ruiz-de-Villa** [1], **Luis Urrutia-Becerra** [2], **Carla Jara** [2], **Mariona Gil i Cortiella** [3], **Joan Miquel Canals** [4], **Albert Mas** [5,*], **Cristina Reguant** [2] **and Nicolas Rozès** [1]

1. Grup de Biotecnologia Microbiana dels Aliments, Departament de Bioquímica i Biotecnologia, Facultat d'Enologia, Universitat Rovira i Virgili, C/Marcel·lí Domingo s/n, 43007 Tarragona, Spain; candela.ruiz@urv.cat (C.R.-d.-V.); nicolasrozes@urv.cat (N.R.)
2. Department of Agro-Industry and Enology, Facultad de Ciencias Agronómicas, Universidad de Chile, Santa Rosa 11315, Santiago 8820808, Chile; carlajara@u.uchile.cl (C.J.); cristina.reguant@urv.cat (C.R.)
3. Instituto de Ciencias Químicas Aplicadas, Inorganic Chemistry and Molecular Material Center, Facultad de Ingeniería, Universidad Autónoma de Chile, Av. El Llano Subercaseaux 2801, Santiago 8910060, Chile; mariona.gil@uautonoma.cl
4. Grup de Tecnologia Enològica, Departament de Bioquímica i Biotecnologia, Facultat d'Enologia, Universitat Rovira i Virgili, C/Marcel·lí Domingo s/n, 43007 Tarragona, Spain
5. Grup de Biotecnologia Enològica, Departament de Bioquímica i Biotecnologia, Facultat d'Enologia, Universitat Rovira i Virgili, C/Marcel·lí Domingo s/n, 43007 Tarragona, Spain
* Correspondence: albert.mas@urv.cat

**Abstract:** Climate change is posing a major challenge to the wine industry, with rising alcohol levels emerging as an issue of concern affecting quality, economics and health. This study explores two methods to reduce alcohol content in Chardonnay wines from Chile. Firstly, 5% and 10% of water was added to grape must. Secondly, the sequential inoculation of *Metschnikowia pulcherrima* with *Saccharomyces cerevisiae* was examined. The main objectives were to assess the efficacy of these treatments in reducing alcohol levels and their impact on organoleptic properties. Our findings revealed that the presence of *M. pulcherrima* in winery conditions was less effective in reducing ethanol. Nevertheless, wines resulting from this treatment exhibited an interesting composition with distinct sensory profiles. Furthermore, the Sc-5% W condition displayed promising results by reducing ethanol content by 0.47% ($v/v$), with less significant changes in the sensory profile. Although the Sc-10% W wines showed a more substantial ethanol reduction of 1.73% ($v/v$), they exhibited a decreasing trend in volatile compounds and polysaccharides, ultimately being perceived as less complex in sensory analysis and not being preferred by consumers. This research contributes to understanding how these approaches affect the alcohol content and sensory attributes of white wines and is fundamental to the sustainability of the sector and the ability of the sector to recover from climate challenges.

**Keywords:** watering; low-alcohol wines; non-*Saccharomyces*

## 1. Introduction

In the last decades, climate change has had a significant and profound impact on the agricultural industry. Therefore, viticulture and winemaking are being affected by this issue in wine regions around the world [1–3]. While climate change is causing challenges and negative effects in traditional winemaking regions, it has also resulted in the emergence of new wine-producing areas due to the displacement of climate patterns [4,5]. Chilean wine regions are also grappling with global warming. Coquimbo, Aconcagua and Central Valley Regions have undergone a change from warm to hot climates [5]. It is worth noting that the Maule region, which provided the grape must for the study, is part of the Central

Valley region. Worldwide global warming produces an increase in temperatures and a reduction in water availability due to draught. In a vineyard, this leads to increased sugar concentrations in grape berries, a reduction in the total acidity levels in grapes and a lag between phenolic and technological maturity [1]. As a result, the wines produced present an organoleptic imbalance with higher alcohol contents and an increased risk of stuck fermentations [1,2].

Furthermore, it is important to note that the increase in alcohol content is related to higher taxes, as a result of policy interventions implemented in numerous countries [6]. Another factor contributing to the reduction in alcohol degree is the growing trend towards adopting healthier lifestyles that involve consuming less alcohol [7].

A wide variety of practices have been proposed to address the issue of increased alcohol content in wines. Many of these approaches focus on viticulture practices, such as modifying irrigation techniques or adjusting pruning management [1]. However, there is also growing research into microbiological modulation to reduce alcohol levels in wines. While carbon metabolic pathways in yeast species are generally conserved, variations exist in terms of ethanol yields [8,9]. Studies have demonstrated that certain non-*Saccharomyces* yeasts can effectively reduce alcohol content through co-fermentation with *Saccharomyces cerevisiae* under aeration conditions [8,10]. Among these yeasts, *Metschnikowia pulcherrima* has been reported as one of the most effective species reducing ethanol levels through its respiratory catabolism of sugars [9–11]. Physical practices have been also proposed as osmotic distillation, reverse osmosis or vacuum distillation [12,13]. Another oenological practice studied to reduce alcoholic content is water addition or substitution in grape must. A reduction between 0.6% *v/v* and 5.9% *v/v* has been reported in the literature, and most of these studies were performed on red wines [14–17]. However, it is worth noting that Gardner et al. [18] conducted a water addition study with the Viognier and Marsanne grape cultivars. The addition of water is not authorized in all wine regions due to varying legislation. For instance, in the state of California (not below 22° Brix) and Australia (not below 24° Brix), the addition of water has been permitted to facilitate the alcoholic fermentation (AF) of must with high sugar content. However, in the European Union, South Africa and other wine-growing regions, the addition of water is generally prohibited unless necessary for additives [19]. In Chile, water addition is authorized in musts with higher levels of soluble solids up to 23.5° Brix, but it is limited to a maximum of 3.5% of the total water allowed for the addition of additives [20].

These findings highlight the potential for exploring alternative practices to mitigate the increase in alcohol content in wines. Furthermore, there is a lack of studies that focus on these practices in white wines, making the research on the Chardonnay cultivar particularly interesting due to its high grape berry maturity [21].

The objective of this study is to compare the effectiveness of reducing the alcohol content of a wine by adding water (5 and 10%) with using a microbiological treatment, *M. pulcherrima* and *S. cerevisiae* sequential fermentations, under pilot-plant-like conditions. We performed an assessment of the physicochemical and sensory properties of the wines with the end goal to determine the most effective method of alcohol reduction and to understand its impact on the final product.

## 2. Materials and Methods

### 2.1. Microorganisms and Alcoholic Fermentation

Two yeast species were evaluated: *S. cerevisiae* QA23 (Sc) for the control and water addition conditions and *M. pulcherrima* Level 2 Flavia (Mp), both from Lallemand Inc., Montreal, BC, Canada. The yeasts were inoculated from dry active yeast and rehydrated according to the manufacturer's instructions. *S. cerevisiae* strains were rehydrated at 37 °C for 30 min, while *M. pulcherrima* strains were rehydrated at 30 °C for the same duration. *S. cerevisiae* was inoculated at a concentration of $2 \times 10^6$ cell/mL and *M. pulcherrima* at a concentration of $10^7$ cell/mL.

The fermentations were conducted using Chardonnay natural grape must supplied by Viña Correa Albano from the Maule region, Chile. Pilot-plant-scale fermentations were performed in 20 L food-grade plastic tanks, maintaining a temperature of 16 °C without agitation. Two different watering conditions were tested by adding distilled water to the grape must: 5% and 10% (which will be referred to in this text as Sc-5% W and Sc-10% W, respectively). In addition to the watering conditions, a sequential fermentation was carried out. *M. pulcherrima* (Mp + Sc) was initially inoculated, and it remained in grape must for 3 days at 22 °C with a manual aeration three times a day. Then, *S. cerevisiae* was inoculated and continued the AF at 16 °C without aeration. Therefore, four conditions were studied in triplicate: Sc-Control (*S. cerevisiae* under control conditions), Sc-5% W (*S. cerevisiae* under 5% watering condition), Sc-10% W (*S. cerevisiae* under 10% watering condition) and Mp + Sc (sequential fermentation with *M. pulcherrima* followed by *S. cerevisiae*). To ensure proper nutrition for yeast growth, at 48 h after the inoculation, nutrients (Nutrienvit, Lallemand Inc., Montreal, Canada) were added to the fermentations at a concentration of 150 mg/L following suppliers' indications.

Inocula and population dynamics were determined by plating a 1:10 serial dilution in YPD agar (10 g/L of yeast extract, 20 g/L of peptone, 20 g/L of glucose, 17 g/L of agar, Panreac Química SLU, Castellar del Vallés, Spain). In addition, populations of *M. pulcherrima* and non-*Saccharomyces* were controlled by Wallerstein selective medium (BDDifco, Billerica, MA, USA).

The monitoring of AF was performed by measuring density each day with "Densito 30PX Portable Density Meter" (Mettler Toledo, Galdakao, Spain). Fermentation was considered finished when the density remained stable and the reducing sugars concentration was less than 2 g/L. Reducing sugars were analysed following the official method of the OIV [22]. After AF, wines were sulphited (10 mg/L $K_2S_2O_5$) and stabilized at 4 °C. Then, they were bottled and stored before the sensory analysis.

### 2.2. Oenological Parameters

General oenological parameters analysed after AF were the following: titratable acidity (expressed as g of equivalent tartaric acid per litre), volatile acidity (expressed as g of equivalent acetic acid per litre), ethanol contents (% *v/v*) and total polyphenol index ($I_{280}$). They were determined by the official method of the OIV [22].

### 2.3. Analysis of Volatile Compounds

The volatile compounds in the wine samples were analysed using the procedure described in [23]. Briefly, before the analysis, the wine samples underwent a pre-treatment process to extract the volatile compounds using headspace solid-phase microextraction (HS-SPME). A 2 cm 50/30 µm carboxen/divinylbenzene/polydimethylsiloxane (CAR/DVB/PDMS) SPME fibre (Supelco, Bellefonte, PA, USA) was employed. The sample volume used was 7.5 mL, where 10 µL of 4-methyl-2-pentanol (0.75 mg/L), used as an internal standard, was added. Headspace sampling was conducted using an autosampler, with the vial incubated at 45 °C for 20 min and agitated at 500 rpm. Subsequently, injection was performed using the spitless mode for 3 min, with a transfer line temperature of 280 °C. Gas chromatography analysis was conducted using a 7890B Agilent GC system coupled to a quadrupole mass spectrometer Agilent 5977 inert (Agilent Technologies, Palo Alto, CA, USA). A DB Wax capillary column (60 m × 0.25 mm × 0.25 µm film thickness, J&W Scientific, Folsom, CA, USA) was used, and the carrier gas was helium flowing at a rate of 1 mL/min.

Compound identification was performed by using an MS ChemStation (Agilent Technologies, Santa Clara, CA, USA). Data were presented as relative area values. To calculate the relative area, the peak area of the main ion of each compound was divided by the peak area of the main ion of internal standard, normalizing it.

### 2.4. Determination of Soluble Polysaccharides

The analysis of polysaccharides was conducted following the procedure outlined in [24]. Initially, the polysaccharides were extracted from the wine matrix using a precipitation method involving cold acidified ethanol. Then, the determination of polysaccharides was carried out using the HRSEC-RID technique.

### 2.5. Analysis of Low-Molecular-Mass Phenolic Compounds

Low-molecular-mass phenolic compounds were analysed following the procedure described in [25]. Briefly, wine phenolic compounds were extracted three times with 25 mL of diethyl ether and three times with 25 mL of ethyl acetate. Then, the extracts were evaporated under vacuum and dissolved in 2 mL of methanol/water (1:1, *v/v*). Samples were analysed by using an HPLC-DAD (Agilent Technologies, Santa Clara, CA, USA). Finally, identification and quantification were conducted by comparison of their spectra and retention times with external standards (Sigma Aldrich, Santiago de Chile, Chile).

### 2.6. Sensory Analysis

Three different sensory evaluations were conducted to assess the wines. Each glass contained 50 mL of wine for all the analysis.

The first evaluation was a triangle sensory analysis, where a panel of 30 tasters, consisting of trained experts, compared the treatments with the Sc-Control to determine significant differences in a binomial test. To eliminate visual subjectivity, the wines were served in dark glasses and labelled with random 3-digit codes.

The second evaluation involved a descriptive analysis, aiming to provide a detailed sensory profile of the samples that showed significant differences in the triangular sensory analysis. Transparent glasses were used to evaluate the wines. A panel of 10 trained and expert tasters performed this analysis. The tasters used a 15 cm unstructured scale to rank the intensity of various attributes, including colour intensity, aroma intensity, mouthfeel intensity, compote aroma, tropical aroma, stone fruit aroma, floral aroma, lactic aroma, acidity, unctuosity, bitterness and persistence.

Lastly, a consumer preference evaluation was conducted, involving 75 consumers. The purpose was to compare the Sc-10% W treatment wine with the Sc-Control wine. Samples were presented following a Latin square design. The panel involved 27 females, 45 males and 3 nonbinaries, with an average age of 24.6 years old. All the consumers were students and staff from the Faculty of Agronomy (University of Chile).

### 2.7. Statistical Analysis

All conditions were performed in triplicate biological samples. The statistical software used was XLSTAT version 2022.5.1 (Addinsoft, Paris, France). The data were analysed with two-way ANOVA with a post hoc Tukey test (honestly significant difference) with a confidence interval of 95% and significant results with a *p*-value < 0.05. PLS-DA was used to discriminate samples regarding volatile compounds.

Descriptive sensory analyses were conducted using the software Panel Check (V1.4.2 2012), applying an ANOVA test with a significance level of 95% and utilizing the least significant differences (LSD) test for post hoc comparisons. Consumers' paired preference analyses were assessed using the binomial distribution and the probability was calculated according to Golden et al. [26].

## 3. Results

### 3.1. Alcoholic Fermentation Kinetics

There were notable differences in the fermentation times among different conditions in terms of alcoholic fermentation (AF) kinetics, considering the time needed to consume 50% of sugars (T50%) and the maximal consumption rate, as it is shown in Figure 1 and Supplementary Data Table S1. Usually, non-*Saccharomyces* species tend to prolong the AF process due to nutrient competition [27]. However, in the case of Mp + Sc wines, the

AF duration was faster compared to the other conditions (Supplementary Data Table S1). This behaviour can be attributed the presence of *S. cerevisiae* and non-*Saccharomyces* yeasts (Supplementary Data, Figure S1) in the grape must which started the fermentation. The sugar consumption was higher than in the other conditions, which may be due to the temperature change (22 °C) during the three-day contact period of *M. pulcherrima* with the grape must, as well as the manual aeration three times a day. These conditions were implemented to optimize the *M. pulcherrima* respiratory metabolism [28] and improve the ethanol reduction effectiveness.

Among the water addition conditions, it is noteworthy that Sc-10% W followed by Sc-5% W exhibited a faster AF compared to Sc-Control, despite the initial lower density resulting from dilution effects.

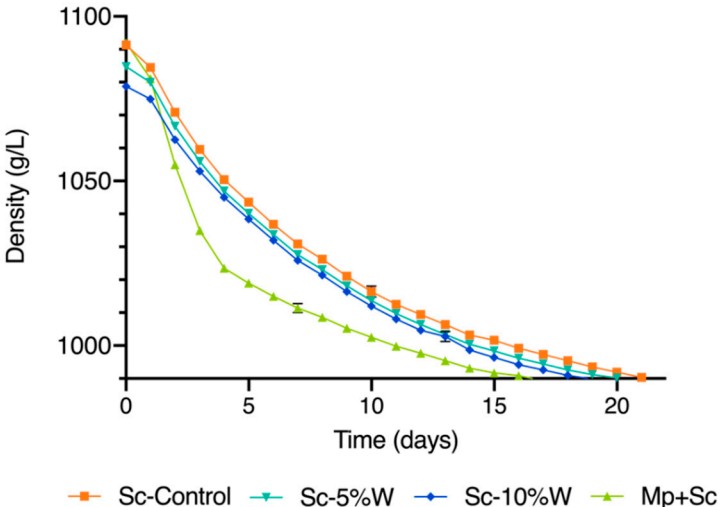

**Figure 1.** Alcoholic fermentation kinetics of the different experimental conditions. The Sc-Control represents wines fermented solely with *S. cerevisiae*, while Sc-5% W and Sc-10% W indicate wines with a pre-fermentative water addition of 5% and 10%, respectively. The Mp + Sc wines depict sequential fermentation with *M. pulcherrima* and *S. cerevisiae*. Means accompanied by standard deviations (SD) based on three replicates (n = 3).

### 3.2. Chemical General Analysis

In terms of reducing alcohol content (Table 1), the Mp + Sc condition showed a tendency towards an ethanol reduction of approximately 0.30% (*v/v*), although these differences were not statistically significant from Sc-Control. Previous studies conducted on natural white must reported alcohol reductions of up to 0.99% (*v/v*) [29], ranging from 0.6% to 1.2% (*v/v*) in Chardonnay must [9] or 0.84% to 1.25% (*v/v*) in the case of the Malvar cultivar [30]. In Muscat wines, Zhu et al. [31] described a reduction up to 0.74% (*v/v*), using the same strain combination as in the present study. It is important to note all these studies involved the sterilization of grape must. However, in our study, we aimed to replicate real semi-industrial vinification conditions, and therefore, we decided not to sterilize the fermenting must. As a result, there was naturally occurring spontaneous yeast present at the beginning of alcoholic fermentation (Supplementary Data, Figure S1). Consequently, the competition between *M. pulcherrima* and the spontaneous yeast [32] could explain the relatively low ethanol reduction observed. Additionally, the differences observed between strains in the literature [9,31] suggested that the compatibility between *M. pulcherrima* and *S. cerevisiae* strains may influence the effectiveness of alcohol reduction.

**Table 1.** Main oenological parameters of final wines. The Sc-Control represents wines fermented solely with *S. cerevisiae*, while Sc-5% W and Sc-10% W indicate wines with a pre-fermentative water addition of 5% and 10%, respectively. The Mp + Sc wines depict sequential fermentation with *M. pulcherrima* and *S. cerevisiae*.

|  | Sc-Control | Sc-5% W | Sc-10% W | Mp + Sc |
|---|---|---|---|---|
| Ethanol % (*v/v*) | 13.93 ± 0.21 [c] | 13.53 ± 0.15 [b] | 12.20 ± 0.10 [a] | 13.67 ± 0.06 [bc] |
| Titratable acidity (g/L de $T_2H$) | 5.12 ± 0.34 | 4,91± 0.32 | 5.19 ± 0.34 | 5.16 ± 0.37 |
| pH | 3.21 ± 0.01 | 3.19 ± 0.01 | 3.21 ± 0.02 [a] | 3.19 ± 0.01 |
| Volatile acidity (g/L) | 0.69 ± 0.03 | 0.68 ± 0.03 | 0.62 ± 0.09 | 0.64 ± 0.03 |
| Reducing sugars (g/L) | 1.87 ± 0.06 [ab] | 1.82 ± 0.26 [ab] | 1.97 ± 0.12 [b] | 1.30 ± 0.37 [a] |
| $I_{280}$ | 9.05 ± 0.36 [c] | 7.25 ± 0.93 [ab] | 7.71 ± 0.31 [a] | 8.95 ± 0.54 [bc] |

Different lowercase letters indicate the existence of significant difference between the samples ($p < 0.05$). Data are expressed as the mean of three experimental replicates ± standard deviation.

On the other hand, the water addition methods managed a better reduction in ethanol content. Sc-5% W resulted in 0.47% (*v/v*) and Sc-10% W showed a reduction of 1.73% (*v/v*), both significantly lower compared to Sc-Control (Table 1). Schelezki et al. [16] reported a reduction of 1% (*v/v*) with a similar water addition in Shiraz musts than in our study (11.6% for early harvest and 10.2% (*v/v*) for late harvest). The addition of water has been tested in other red wines with other percentages. For instance, an addition of 7.5% of water decreased ethanol content by 0.9% (*v/v*) in Shiraz wines [16], an addition of 14% *v/v* resulted in a reduction of 2.1% (*v/v*) also in Shiraz wines [17], and an addition of 8% of water decreased the ethanol content by 1.1% *v/v* in Tempranillo wines [14].

The results suggest that the effectiveness of ethanol reduction can depend on the fermentation approaches and grapes cultivars [19].

Regarding general chemical parameters as titratable acidity, pH or volatile acidity, there were no significant differences among treatments. Regarding titratable acidity, a decrease was anticipated due to the dilution effect caused by the addition of water. However, it is worth noting that the concentration of titratable acidity could be lower in the Sc-Control and Mp + Sc treatments due to tartrate precipitation. Tartaric acid exhibits higher insolubility in ethanol, which means that as the ethanol content increases, the precipitation of tartrates becomes more pronounced compared to treatments with water addition. This way, the reduction in acidity by tartaric precipitation in Sc-Control and Mp + Sc could be compensated by a dilution effect in treatments with water addition, explaining the lack of statistical differences among treatments.

Nevertheless, $I_{280}$ suffered the dilution effect of grape must, being significantly higher in Sc-Control and Mp + Sc than in water treatments (Table 1). This behaviour, as expected, has been reported previously in water addition studies [15].

### 3.3. Volatile Compounds in Wines

The relative abundance of the identified 57 volatile compounds was analysed (Supplementary Data Table S3). As it was shown on Table 2, the volatile compounds detected were grouped in following families: acetate esters (7), ethyl esters (19), other esters (2), short-chain fatty acids (SCFA) (3) (acetic acid, isobutyric acid and isovaleric acid), medium-chain fatty acids (MCFA) (3) (octanoic acid, decanoic acid and dodecanoic acid), fusel alcohols (10), aldehydes (2), ketones (2) and terpenes (8).

The first two latent variables (LV) of PLS-DA (partial least squares-discriminant analysis) LV1 and LV2 explained 58.48% of the total variance of the **Y**-block (classes of treatments) (Figure 2A). Thus, LV1 was effective in discriminating between wines treated with Sc-control and water and those produced through sequential fermentation (Mp + Sc), which were separated. Most of the volatile compounds were positively correlated with the Mp + Sc treatment (Figure 2B). On the other hand, LV2 helped in differentiating Sc-control wines from Sc-10% W wines, while Sc-5% W wines occupied a middle position between them. In this case, some esters, terpenes and MCFA were positively correlated

with Sc-Control and Sc-5% W. On the contrary, compounds appear negatively correlated with Sc-10% W.

**Table 2.** Volatile compounds expressed as relative abundance. The Sc-Control represents wines fermented solely with *S. cerevisiae*, while Sc-5% W and Sc-10% W indicate wines with a pre-fermentative water addition of 5% and 10%, respectively. The Mp + Sc wines depict sequential fermentation with *M. pulcherrima* and *S. cerevisiae*.

|  | Sc-Control | Sc-5% W | Sc-10% W | Mp + Sc |
|---|---|---|---|---|
| $\sum$ **Acetate esters** | 22.36 ± 0.71 [b] | 17.88 ± 0.32 [a] | 17.39 ± 1.04 [a] | 21.32 ± 1.49 [b] |
| $\sum$ **Ethyl esters** | 60.02 ± 2.65 [c] | 44.06 ± 4.39 [ab] | 40.06 ± 2.83 [a] | 49.05 ± 2.61 [b] |
| $\sum$ **Other esters** | 1.02 ± 0.17 [a] | 1.25 ± 0.10 [a] | 1.04 ± 0.12 [a] | 1.51 ± 0.05 [b] |
| $\sum$ **Total esters** | 83.32 ± 3.41 [c] | 63.08 ± 4.55 [a] | 58.41 ± 3.96 [a] | 71.65 ± 2.93 [b] |
| $\sum$ **SCFA** | 0.21 ± 0.02 | 0.21 ± 0.04 | 0.16 ± 0.01 | 0.29 ± 0.06 |
| $\sum$ **MCFA** | 13.62 ± 0.99 [b] | 13.89 ± 1.92 [b] | 12.96 ± 1.53 [ab] | 9.65 ± 0.45 [a] |
| $\sum$ **Total acids** | 14.49 ± 1.09 [a] | 14.56 ± 1.94 [a] | 13.49 ± 1.52 [a] | 10.56 ± 0.52 [b] |
| $\sum$ **Fusel alcohols** | 23.81 ± 0.89 [a] | 23.44 ± 1.57 [a] | 23.08 ± 1.04 [a] | 33.37 ± 3.04 [b] |
| $\sum$ **Aldehydes** | 0.19 ± 0.01 [a] | 0.19 ± 0.02 [a] | 0.28 ± 0.02 [a] | 0.35 ± 0.08 [b] |
| $\sum$ **Ketones** | 0.14 ± 0.03 | 0.09 ± 0.03 | 0.10 ± 0.01 | 0.10 ± 0.01 |
| $\sum$ **Terpenes** | 0.67 ± 0.07 [b] | 0.59 ± 0.05 [ab] | 0.64 ± 0.05 [a] | 0.77 ± 0.07 [ab] |

Different lowercase letters indicate the existence of significant difference between the samples ($p < 0.05$). Data are expressed as the mean of three experimental replicates ± standard deviation.

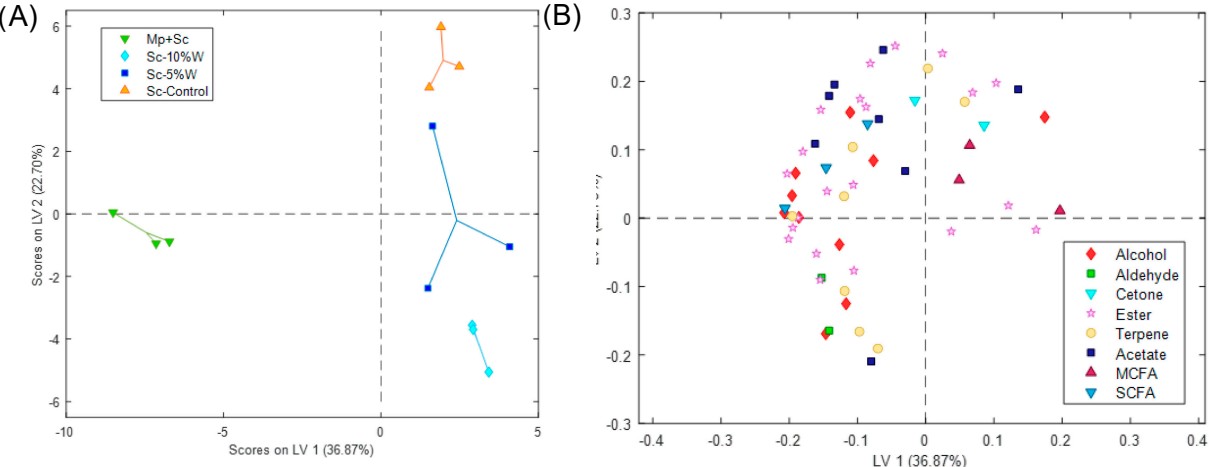

**Figure 2.** First two latent variables for the PLS-DA model (58.48% of the variance) of the volatile compounds. (**A**) Scores and (**B**) loadings on the PLS-DA model. The Sc-Control represents wines fermented solely with *S. cerevisiae*, while Sc-5% W and Sc-10% W indicate wines with a pre-fermentative water addition of 5% and 10%, respectively. The Mp + Sc wines depict sequential fermentation with *M. pulcherrima* and *S. cerevisiae*. Means accompanied by standard deviations (SD) based on three replicates (n = 3).

Regarding the relative abundance of volatile families, there were some interesting differences observed (Table 2). The presence of *M. pulcherrima* during AF appeared to increase the overall levels of fusel alcohols, which contributes to a floral aroma. This increase in higher alcohols has been previously reported in white wines fermented with *M. pulcherrima* in sequential culture with *S. cerevisiae* [9,29,30,33]. Specifically, the presence of 2-phenyletanol in sequential fermentation with non-*Saccharomyces* yeasts has been attributed to *M. pulcherrima* [34]. Hranilovic et al. [9] proposed that this effect could be linked to a response from sequential inoculation. Wines fermented with a combination of *M.*

*pulcherrima* and *S. cerevisiae* (Mp + Sc) also exhibited an increase in total aldehydes, which contributes to a fruity aroma. Furthermore, certain terpenes showed increased levels in Mp + Sc wines. This could be associated with the higher activity of β-glucosidase, activity described in *M. pulcherrima* [35]. In contrast, the abundance of ethyl esters was reduced in Mp + Sc wines, which agrees with Tronchoni et al. [36], who described a higher presence of ethyl esters in wines fermented with *S. cerevisiae* monocultures compared with wines in sequential fermentations with *M. pulcherrima*. However, other studies have reported no variation or an increase in ethyl esters with the presence of *M. pulcherrima* [9,29]. These findings suggest that the production of ethyl esters associated with *M. pulcherrima* may be influenced by the winemaking conditions. Furthermore, the abundance of MCFA was significantly reduced in Mp + Sc wines in accordance with several reports [9,29,36]. Specifically, Balmaseda et al. [37] observed a higher reduction in the white grape variety Macabeo compared to Cabernet sauvignon, due to the differences in the vinification process. This decrease is noteworthy because these compounds are known to be toxic to *Oenococcus oeni* and consequently reduce malolactic activity [38,39].

In the context of pre-fermentative water treatments, it has been observed that the reduction in volatile compounds varies among different families. Notably, total esters show a significant decrease when subjected to water treatments. This dilution effect in water addition was previously described by other researchers [16,18]. However, other volatile families such as acids, fusel alcohols or aldehydes had no change in relation to the Sc-control. Moreover, it is worth highlighting that, apart from total terpenes, there are no significant differences observed between the two dilution percentages, Sc-5% W and Sc-10% W. This suggests that the effect of dilution with water prior to the AF on wine volatile compounds is not strongly influenced by the dilution percentage within this range.

### 3.4. Soluble Polysaccharides in Wines

The method employed in this study enabled the identification of four distinct fractions containing polysaccharides from grapes and microorganisms (yeasts and bacteria) [24]. Figure 3 illustrates these different fractions detected, each corresponding to a specific molecular weight range: the high-molecular-weight fraction (HMWf) with a number average molecular weight Mn of $158.7 \pm 2.4$ KDa, the medium-molecular-weight fraction (MMWf) with an Mn = $34.3 \pm 0.6$ KDa, the low-molecular-weight fraction (LMWf) with an Mn = $16.3 \pm 0.6$ KDa and the oligosaccharide fraction (OLIGf) with an Mn = $5.9 \pm 0.2$ KDa.

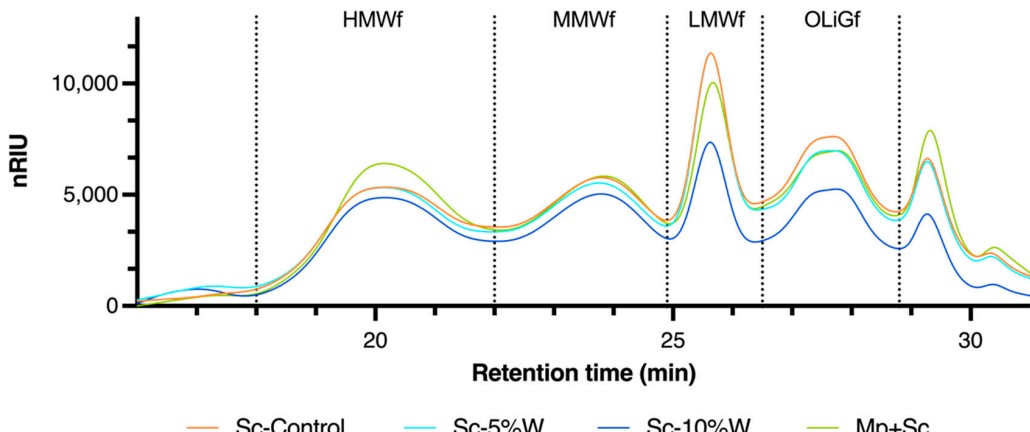

**Figure 3.** Molecular weight distribution of soluble polysaccharides fractions, by HRSEC-RID. The Sc-Control represents wines fermented solely with *S. cerevisiae*, while Sc-5% W and Sc-10% W indicate wines with a pre-fermentative water addition of 5% and 10%, respectively. The Mp + Sc wines depict sequential fermentation with *M. pulcherrima* and *S. cerevisiae*.

In terms of total concentration, significant differences were observed among the Chardonnay wines examined in this study. Sequential fermentation with *M. pulcherrima*

resulted in notably higher concentrations of total polysaccharides. While slight differences were observed for MMWf and OLIGf, a higher increment was found in the HMWf fraction, which exhibited a 20.45% increase in Mp + Sc wines compared to the Sc-Control wine (Table 3). Previous studies have reported that non-*Saccharomyces* yeasts release more polysaccharides during AF, which are essentially mannoproteins [40–42]. González-Royo et al. [43] concluded that the presence of *M. pulcherrima* in white wine fermentation leads to an overall increase in total polysaccharides, with the most substantial increase observed in the HMWf fraction (Figure 3).

**Table 3.** Polysaccharide fractions (mg/L). The Sc-Control represents wines fermented solely with *S. cerevisiae*, while Sc-5% W and Sc-10% W indicate wines with a pre-fermentative water addition of 5% and 10%, respectively. The Mp + Sc wines depict sequential fermentation with *M. pulcherrima* and *S. cerevisiae*.

| Fraction (mg/L) | Sc-Control | Sc-5% W | Sc-10% W | Mp + Sc |
|---|---|---|---|---|
| **HMWf** | 43.33 ± 3.02 [a] | 44.05 ± 2.35 [a] | 40.81 ± 4.06 [a] | 52.19 ± 0.67 [b] |
| **MMWf** | 37.09 ± 1.74 [bc] | 35.31 ± 0.78 [ab] | 33.08 ± 3.07 [a] | 40.29 ± 0.18 [c] |
| **LMWf** | 31.62 ± 1.06 [b] | 20.42 ± 4.36 [a] | 18.43 ± 1.90 [a] | 31.58 ± 5.16 [b] |
| **OLIGf** | 41.25 ± 1.03 [c] | 34.10 ± 0.16 [b] | 27.76 ± 2.55 [a] | 43.23 ± 3.01 [c] |
| ∑ **Polysaccharides** | 153.29 ± 5.74 [b] | 133.89 ± 5.62 [a] | 127.61 ± 7.78 [a] | 167.28 ± 6.07 [b] |

Different lowercase letters indicate the existence of significant difference between the samples ($p < 0.05$). Data are expressed as the mean of three experimental replicates ± standard deviation.

When observing the effects of water addition treatments, it was found that polysaccharides experienced a significant reduction. However, the fractions that exhibited the most significant variations were MMWf, LMWf and OLIGf (Table 3).

Moreover, noteworthy differences were observed between the different percentages of water addition (Figure 3). In the case of MMWf, a non-significant reduction of 4.8% was observed in Sc-5% W wines, while a larger reduction of 10.8% was observed in Sc-10% W wines. In the LMWf fraction, a 35.4% reduction was observed in Sc-5% W, compared to a higher reduction of 41.7% in Sc-10% W wines. Lastly, the OLIGf fraction experienced a 17.3% reduction in Sc-5% W, whereas a more significant reduction of 32.7% was observed with a 10% water addition (Table 3). Piccardo et al., Schelezki et al. and Teng et al. [14,17,44] also described a reduction in the content of polysaccharides in red wines with the addition of water.

*3.5. Low-Molecular-Mass Phenolic Compounds in Wines*

The present study investigated the impact of different treatments on the phenolic composition of wines. Various phenolic compounds were identified, including hydroxybenzoic acids (gallic acid and protocatechuic acid), hydroxycinnamic acids and derivatives (trans-caftaric acid, trans-coutaric acid, cis-coutaric acid, caffeic acid and hexose ester of trans *p*-coumaric acid), phenolic alcohols (tyrosol), flavanols (catechin, epicatechin and procyanidins) and flavonols (astilbin, quercetin, derivatives and other flavanols).

The heatmap presented in Figure 4 illustrates the proportions of phenol compounds between different treatments. In the case of Mp + Sc wines, there was a tendency towards increased levels of flavanols, hydroxybenzoic acids and hydroxycinnamic acids. Conversely, a decrease was observed in flavonols and phenolic alcohols, although these differences did not reach statistical significance. However, two notable significant variations were observed. The content of epicatechin increased from 8.30 mg/L in the Sc-Control wine to 9.47 mg/L in the presence of *M. pulcherrima*. On the other hand, quercetin levels decreased from 0.92 mg/L in the Sc-Control wine to 0.61 mg/L in Mp + Sc wines (Supplementary Table S2).

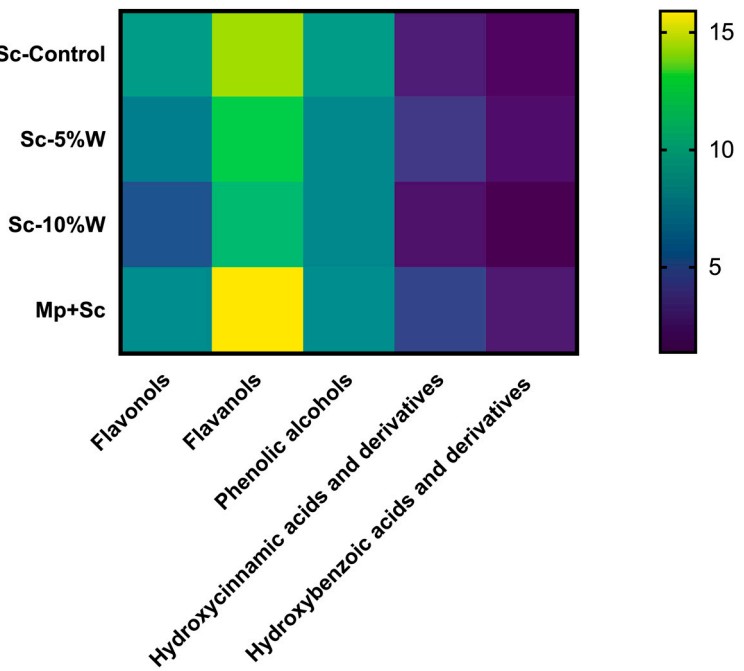

**Figure 4.** Heat map of low-molecular-mass phenolic compounds families detected in wines. The Sc-Control represents wines fermented solely with *S. cerevisiae*, while Sc-5% W and Sc-10% W indicate wines with a pre-fermentative water addition of 5% and 10%, respectively. The Mp + Sc wines depict sequential fermentation with *M. pulcherrima* and *S. cerevisiae*.

There have been relatively few studies analysing the modification of low-molecular-mass phenolic compounds in white wines with the presence of *M. pulcherrima*. However, it has been described that *M. pulcherrima* can increase polyphenolic content in red wines [37,45]. In addition, Sorrentino et.al [46] reported an increase in certain phenolic compounds such as epicatechin, catechin and gallic acid, with the use of another species of Metschnikowia genera: *M. fruticola*. This increase has been attributed to higher polygalacturonase activity [47], as suggested by some authors, although no statistical differences were observed for wine soluble polysaccharides, which should be also related to enzymatic activity through maturity and AF.

Regarding the Sc-5% W water treatment, there was a general decreasing trend observed in all low-molecular-mass compounds, although no significant differences were found among the phenolic compound families (Figure 4). Nevertheless, specific phenols such as epicatechin, quercetin and certain procyanidins showed reductions (Supplementary Table S2). On the contrary, Sc-10% W wines exhibited lower values across all families of compounds (Figure 4), indicating a dilution effect like what has been observed in some volatile compounds and polysaccharides. These results would agree with the reported effects of water additions to the fermenting must, although it was observed in red wines [14,15,17].

### 3.6. Sensory Analysis

A triangular sensory analysis was conducted to assess the differences between the control and the treatments. The results showed that out of 30 tasters, 21 were able to differentiate the Mp + Sc wine from the Sc-Control, 14 were able to differentiate the Sc-10% W wine, and 12 were able to differentiate the Sc-5% W wine from the Sc-Control. Based on these results, only the Mp + Sc wine and Sc-10% W wine were found to be significantly different from the Sc-Control wine, with a significance level of $p < 0.1$. Following the triangular analysis, a descriptive analysis was performed by a professional tasting panel to further evaluate the sensory differences between the significantly different wines. The treatments compared in this analysis were the Sc-Control, Sc-10% W and Mp + Sc, as depicted in Figure 4.

Figure 5A,B indicate that there were only significant differences between colour intensity, acidity and unctuosity. The Sc-10% W wines had significantly less unctuosity and colour intensity, suggesting that the addition of water clearly influences the decrease in unctuosity. On the contrary, the acidity perception was not the higher, as could be expected. Mp + Sc wines were rated as the most acidic in comparison to the Sc-Control; even the analytical value of total acidity was not different. Regarding aromas, although no significant differences were found between the conditions, there was a decreasing trend in aroma intensity for Sc-10% W wines, which aligns with the reduction in volatile compounds mentioned above. Other researchers studying red wines with added water also reported a decrease in flavour intensity, colour and structural characteristics, which became more pronounced with higher water addition percentages; however, the tannin levels remained stable [15,16]. Substitution methods had fewer effects on the sensory profiles. Piccardo et al. [14] described wines with the addition or substitution of water as having more vegetal and acidic characteristics. In red wines, sensory changes may be more pronounced due to the higher complexity compared to white wines.

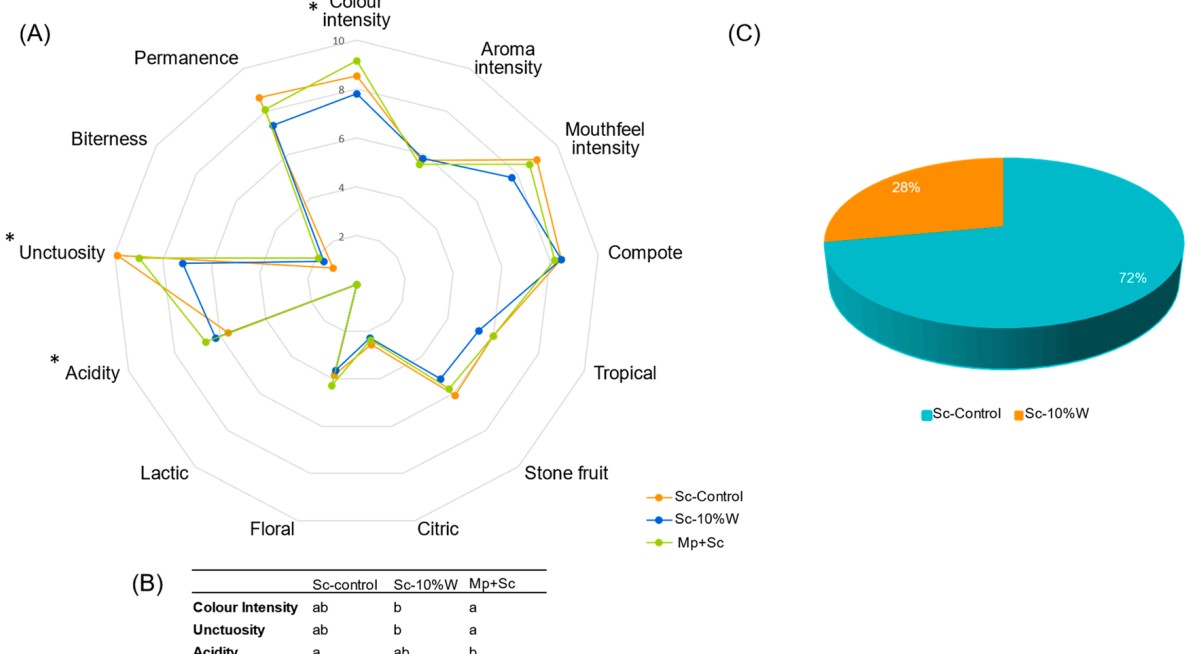

**Figure 5.** (**A**) Spider plot of organoleptic parameters analysed in descriptive analysis. Asterisks (\*) indicate significant differences between conditions: \* $p < 0.05$. (**B**) Letters indicating significant differences ($p < 0.05$) between conditions. (**C**) Percentage of preference in the analysis of consumer preference. The Sc-Control represents wines fermented solely with *S. cerevisiae*, while Sc-10% W indicates wines with a pre-fermentative water addition of 10%. The Mp + Sc wines depict sequential fermentation with *M. pulcherrima* and *S. cerevisiae*.

In wines fermented with the presence of *M. pulcherrima*, the tropical and fruity notes were not significantly perceived, which is consistent with the lower ester concentrations compared to the Sc-Control wines. However, the floral notes were still present, possibly due to a high concentration of some fusel alcohols. In addition, it has been described in other sensory analysis that wines fermented in the presence of *M. pulcherrima* had oxidation and spirit-like aromas [36], which could be attributed to the high levels of isoamyl alcohol.

Finally, to validate the impact of water addition on the wine profile, we conducted a preference test, comparing Sc-Control with Sc-10% W. The results clearly demonstrated a strong preference for the Sc-Control sample without water addition, with a *p*-value of $5.6 \times 10^{-5}$ (Figure 5C). This preference could be attributed to the higher levels of unctuosity observed in the Sc-Control sample. Interestingly, while a previous study by Niimi et al. [48]

indicated that an increase in body did not influence preference in red wine, our findings suggest that consumers associate unctuosity with quality specifically in white wine.

## 4. Conclusions

This study aimed to investigate different methods for reducing alcohol content in wines, including the use of *Metschnikowia pulcherrima* in sequential fermentation with *Saccharomyces cerevisiae* and the addition of water (5% and 10%) to the fermentative must. The experiments focused on Chardonnay wines, as there is a lack of research on these technologies in white wines. In order to simulate real conditions, the fermentative must was not sterilized, in order to study the reduction in ethanol and sensory modifications of treatments under competitive pressure of endogenous microorganisms.

Our findings indicated that the presence of *M. pulcherrima* in alcoholic fermentation was less effective in reducing ethanol, likely due to the presence of other yeast species in the must. However, the resulting wines had different compositions, with higher levels of HMWf polysaccharides and a tendency towards increased concentrations of certain phenolic compounds, particularly epicatechin. In terms of volatile compounds, there was an increase in fusel alcohols, which could be linked to the heightened floral notes of the wines in sensory analysis. Regarding pre-fermentative water addition, the Sc-5% W condition, which reduced ethanol by $0.47 \pm 0.06\%$ (*v/v*), showed promising results in terms of analytical parameters, with no significant differences observed in low-molecular-weight phenolic compounds. This condition also exhibited a slighter reduction trend in volatile compounds and polysaccharides compared to the Sc-10% W wines, which had an ethanol reduction of $1.73 \pm 0.10\%$ (*v/v*). Wines with a pre-fermentative water addition of 10% were described as less complex in sensory analysis, showing a decreasing trend in all analysed organoleptic parameters.

These results demonstrate that adding high percentages of water leads to a general decrease in the concentration of most wine components, although it can also increase wine production, which may pose a challenge. However, the addition of lower percentages of water, such as 5% or even less, can effectively reduce ethanol content without significantly altering the organoleptic profile of the wines. Further research could explore the combination of low water addition percentages with the use of non-*Saccharomyces* yeast, such as *M. pulcherrima*, to achieve both ethanol reduction and improvements in organoleptic characteristics. This line of research is important considering the rapid progress of climate change and the limited approval rates for these methods in most wine-producing countries.

**Supplementary Materials:** The following supporting information can be downloaded at: https://www.mdpi.com/article/10.3390/fermentation9090808/s1, Figure S1: (A) Sc-Control, Sc-5% W and Sc-10% W yeast populations during AF. (B) Mp + Sc yeast populations during AF. The Sc-Control represents wines fermented solely with *S. cerevisiae*, while Sc-5% W and Sc-10% W indicate wines with a pre-fermentative water addition of 5% and 10%, respectively. The Mp + Sc wines depict sequential fermentation with *M. pulcherrima* and *S. cerevisiae*. Means accompanied by standard deviations (SD) based on three replicates (n = 3); Table S1: Maximal Consumption Rate (g/L/day) and days to consume 50% of sugars (T50%). The Sc-Control represents wines fermented solely with *S. cerevisiae*, while Sc-5%W and Sc-10%W indicate wines with a pre-fermentative water addition of 5% and 10%, respectively. The Mp+Sc wines depict sequential fermentation with *M. pulcherrima* and *S. cerevisiae*. Table S2: Low-molecular-mass phenolic compounds in wines (mg/L). The Sc-Control represents wines fermented solely with *S. cerevisiae*, while Sc-5% W and Sc-10% W indicate wines with a pre-fermentative water addition of 5% and 10%, respectively. The Mp + Sc wines depict sequential fermentation with *M. pulcherrima* and *S. cerevisiae*. Table S3: Volatile compounds detected in wines (relative area). The Sc-Control represents wines fermented solely with *S. cerevisiae*, while Sc-5%W and Sc-10%W indicate wines with a pre-fermentative water addition of 5% and 10%, respectively. The Mp + Sc wines depict sequential fermentation with *M. pulcherrima* and *S. cerevisiae*.

**Author Contributions:** Conceptualization, A.M., J.M.C., M.G.i.C., C.J. and C.R.-d.-V.; methodology, A.M., J.M.C., M.G.i.C., C.J. and C.R.-d.-V.; software, C.R.-d.-V. and J.M.C.; validation, A.M., J.M.C., M.G.i.C., C.J. and C.R.-d.-V.; formal analysis, A.M., J.M.C., M.G.i.C. and C.J.; investigation, C.R.-d.-V.

and L.U.-B.; resources, A.M., J.M.C., M.G.i.C. and C.J., data curation, C.R.-d.-V.; writing—original draft preparation, C.R.-d.-V.; writing—review and editing, A.M., J.M.C., M.G.i.C., C.J., N.R. and C.R.; visualization, A.M., J.M.C., M.G.i.C., C.J., N.R. and C.R.; supervision, A.M., J.M.C., M.G.i.C., C.J., N.R. and C.R.; project administration, A.M. and J.M.C.; funding acquisition, A.M. and J.M.C. All authors have read and agreed to the published version of the manuscript.

**Funding:** This work was supported by grant PGC2018-101852-B-I00 awarded by the Spanish Research Agency. This publication was possible with the support of the Secretaria d'Universitats i Recerca del Departament d'Empresa i Coneixement de la Generalitat de Catalunya (2020 FISDU 00359; Ruiz-de-Villa, C.). C. Ruiz-de-Villa and J.M. Canals stage at University of Chile was financially supported by SC-RISE Grant vWISE from the European Union.

**Data Availability Statement:** The data presented in this study are available on request from the corresponding authors.

**Acknowledgments:** The authors would like to thank Alvaro Peña-Neira for the use of the facilities, Marcela Medel-Maraboli and Karinna Estay for their support with sensory analysis and Jokin Ezenarro for his support with Matlab 2022a software and pls toolbox 9.1.

**Conflicts of Interest:** The authors declare that they have no known competing financial interests or personal relationships that could have appeared to influence the work reported in this paper.

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
