# Peer review of "Physicochemical and Organoleptic Differences in Chardonnay Chilean Wines after Ethanol Reduction Practises: Pre-Fermentative Water Addition or Metschnikowia pulcherrima"

_fermentation, doi:10.3390/fermentation9090808_

Round 1

Reviewer 1 Report

- several italics must to be included (abstract, supplementary data)

- figure 1 needs to be analyzed, maybe by using T50 (time needed to consume the 50% of sugars) in order to carry out statistics

-Furthermore, regarding figure 1:

I agree with the use Sc as a control of water addition conditions, however I think Sc would not be the correct control for the sequential inoculation experiment because in this fermentations authors changed temperature, aeration... the correct controls should be Sc and Mp in individual fermentations and using exactly the same conditions as those used in the sequential fermentations otherwise the conclusions can be biased. 

Author Response

Responses to Reviewer 1

- several italics must to be included (abstract, supplementary data)

The italics in all the manuscript have been revised.

- figure 1 needs to be analyzed, maybe by using T50 (time needed to consume the 50% of sugars) in order to carry out statistics

We thank the reviewer for the suggestion. We have calculated the maximal consumption rate and the time needed to consume 50% of sugars to provide statistical analysis. These data are shown in Supplementary data: Table S1.

-Furthermore, regarding figure 1:

I agree with the use Sc as a control of water addition conditions, however I think Sc would not be the correct control for the sequential inoculation experiment because in this fermentations authors changed temperature, aeration... the correct controls should be Sc and Mp in individual fermentations and using exactly the same conditions as those used in the sequential fermentations otherwise the conclusions can be biased. 

We agree with the reviewer about the proper controls for the sequential inoculation. However, this has already been done several times in the past (Contreras et al., 2014; Hranilovic et al., 2020; Zhu et al., 2020) and it is not the main focus of the manuscript. The focus was on the ethanol reduction by physical means, using an external (biological) method for comparison purposes.

Reviewer 2 Report

This study was well designed and conducted and the manuscript is well written.

Comment:

1.     Line112, WL Differential Medium is used for isolating bacteria encountered in brewing and industrial fermentation processes, so I wonder if it can control the populations of M. 111 pulcherrima and non-Saccharomyces.

Some minor points should be corrected

1.      Line 114. The monitoring of AF was performed by measuring density each day with” (“Densito.

2.      Line 100 and Line 196. The M. pulcherrima (Mp+Sc) was initially inoculated, at 24°C or 22 °C?

3.      Line 252. I280 should be defined

Author Response

This study was well designed and conducted and the manuscript is well written.

Comment:

  1. Line112, WL Differential Medium is used for isolating bacteria encountered in brewing and industrial fermentation processes, so I wonder if it can control the populations of 111 pulcherrima and non-Saccharomyces.

Thanks to the reviewer for this comment. WL (Wallerstein Laboratory) Differential Agar medium is used in several studies to differentiate non-Saccharomyces yeast. Some of these works include:

  • Gerard, L. M., Corrado, M. B., Davies, C. V., Soldá, C. A., Dalzotto, M. G., & Esteche, S. (2023). Isolation and identification of native yeasts from the spontaneous fermentation of grape musts. Archives of Microbiology, 205(9), 302. https://doi.org/10.1007/s00203-023-03646-1
  • Polizzotto, G., Barone, E., Ponticello, G., Fasciana, T., Barbera, D., Corona, O., Amore, G., Giammanco, A., & Oliva, D. (2016). Isolation, identification and oenological characterization of non-Saccharomyces yeasts in a Mediterranean island. Letters in Applied Microbiology, 63(2), 131–138. https://doi.org/10.1111/lam.12599

Some minor points should be corrected.

  1. Line 114. The monitoring of AF was performed by measuring density each day with” (“Densito.

Thanks to the reviewer for this comment. The sentence has been corrected.

  1. Line 100 and Line 196. The M. pulcherrima (Mp+Sc) was initially inoculated, at 24°C or 22 °C?

Thanks to the reviewer for this comment. The temperature has been corrected.

  1. Line 252. I280 should be defined

Thanks to the reviewer for this comment. The parameter has been defined in materials and methods.

Reviewer 3 Report

This work provides an detailed investigation on the influence of adding water or Metschnikowia pulcherrima on the physicochemical and organoleptic properties of Chardonnay wines from Chile, and a promising way for reducing the ethanol content of the wine was proposed. The experiment was well performed, and the results were presented logically and clearly. However, there are some minor points to improve the manuscript.

1.       Line 200-202: According to Figure 1, the slope of Sc-10% seems to be not significantly higher than that of Sc-5%, and the slope of Sc-5% seems to be not significantly higher than that of Sc-control. Please change the description if there is no comparison between the slopes.

2.       Line 454-455: This research has no experiment for comparing the reducing ethanol effects with laboratory conditions, so there is no evidence to support the statement. In this part, the authors just need to show their findings on the basis of the present experiments. Similar problem also exists in abstract.

3.       Please provide the data of each individual volatile compound as supplementary materials.

Some grammatical mistakes should be checked and corrected or revised. For example:

Line 42-43: Increased in temperatures and … lead to …, reduced the total acidity levels …

Line 81: non-Saccharomyces yeasts. ?

Line 81-82: Therefore, … Thus, …

Line 399: In red wines has also been reported a decrease in …

Author Response

Responses to Reviewer 2:

Comments and Suggestions for Authors

This work provides a detailed investigation on the influence of adding water or Metschnikowia pulcherrima on the physicochemical and organoleptic properties of Chardonnay wines from Chile, and a promising way for reducing the ethanol content of the wine was proposed. The experiment was well performed, and the results were presented logically and clearly. However, there are some minor points to improve the manuscript.

  1. Line 200-202: According to Figure 1, the slope of Sc-10% seems to be not significantly higher than that of Sc-5%, and the slope of Sc-5% seems to be not significantly higher than that of Sc-control. Please change the description if there is no comparison between the slopes.

We thank the reviewer for the suggestion. In order to provide statistical analysis that support this comment we have calculated the maximal consumption rate and the time needed to consume 50% of sugars. These data are shown in Supplementary data: Table S1.

  1. Line 454-455: This research has no experiment for comparing the reducing ethanol effects with laboratory conditions, so there is no evidence to support the statement. In this part, the authors just need to show their findings on the basis of the present experiments. Similar problem also exists in abstract.

Thanks to the reviewer for the comment. We have eliminate the sentence: “compared to laboratory conditions”.

  1. Please provide the data of each individual volatile compound as supplementary materials.

We appreciate the suggestion. The table has been included (Table S3).

Comments on the Quality of English Language

Some grammatical mistakes should be checked and corrected or revised.  

For example:

Line 42-43: Increased in temperatures and … lead to …, reduced the total acidity levels …

Thanks to the reviewer for the suggestion. We have rephrased the sentence:

Increased in temperatures and reduced water availability lead to increased sugar concentrations in grape berries, reduced the total acidity levels in grapes and a lag between phenolic and technological maturity.

The global warming worldwide produces an increase in temperatures and a reduction of water availability due to draught. In the vineyard this leads to increased sugar concentrations in grape berries, reduced the total acidity levels in grapes and a lag between phenolic and technological maturity.

Line 81: non-Saccharomyces yeasts. ?

Thanks to the reviewer for the suggestion. We have eliminated the sentence.

Line 81-82: Therefore, … Thus, …

We appreciate the comment. The sentence has been rephrased:

Therefore, an assessment of the physicochemical and sensory properties of wines has been conducted. Thus, the end goal is to determine the most effective method of alcohol reduction and understand the impact on the overall quality of the product.

We performed an assessment of the physicochemical and sensory properties of the wines with the end goal to determine the most effective method of alcohol reduction and understand its impact on the final product.

Line 399: In red wines has also been reported a decrease in …

Thanks to the reviewer for this comment. The sentence has been rephrased:

In red wines has also been reported a decrease in polyphenols when water is added to the fermenting must

These results would agree with the reported effects of water additions to the fermenting must, although it was observed in red wines